# Action Dubber: Timing Audible Actions via Inflectional Flow

**Wenlong Wan** [* 1 2]   **Weiying Zheng** [* 3]   **Tianyi Xiang** [* 4 2 1]   **Guiqing Li** [1]   **Shengfeng He** [2]

## Abstract

We introduce the task of Audible Action Temporal Localization, which aims to identify the spatio-temporal coordinates of audible movements. Unlike conventional tasks such as action recognition and temporal action localization, which broadly analyze video content, our task focuses on the distinct kinematic dynamics of audible actions. It is based on the premise that key actions are driven by inflectional movements; for example, collisions that produce sound often involve abrupt changes in motion. To capture this, we propose $TA^2Net$, a novel architecture that estimates inflectional flow using the second derivative of motion to determine collision timings without relying on audio input. $TA^2Net$ also integrates a self-supervised spatial localization strategy during training, combining contrastive learning with spatial analysis. This dual design improves temporal localization accuracy and simultaneously identifies sound sources within video frames. To support this task, we introduce a new benchmark dataset, $Audible623$, derived from Kinetics and UCF101 by removing non-essential vocalization subsets. Extensive experiments confirm the effectiveness of our approach on $Audible623$ and show strong generalizability to other domains, such as repetitive counting and sound source localization. Code and dataset are available at https://github.com/WenlongWan/Audible623.

## 1. Introduction

Videos have become an integral part of daily life on social media platforms, especially on popular video-sharing

*Equal contribution [1]School of Computer Science and Engineering, South China University of Technology [2]School of Computing and Information Systems, Singapore Management University [3]School of Computing and Data Science, University of Hong Kong [4]Department of Computer Science, City University of Hong Kong. Correspondence to: Shengfeng He <shengfenghe@smu.edu.sg>.

*Proceedings of the 42nd International Conference on Machine Learning*, Vancouver, Canada. PMLR 267, 2025. Copyright 2025 by the author(s).

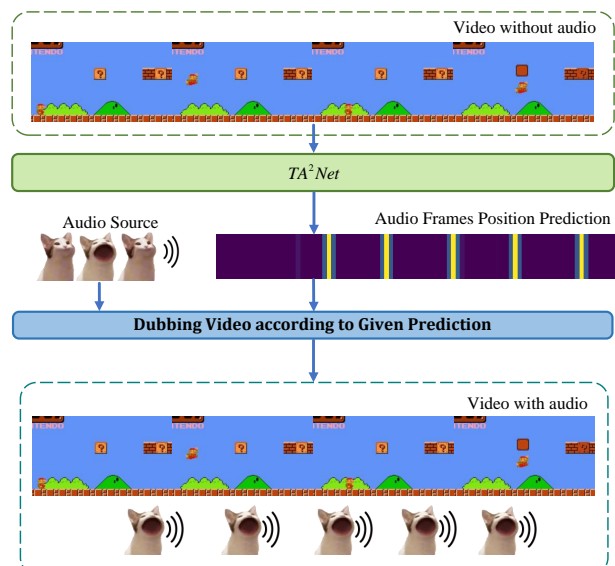

Figure 1: We propose a novel task aimed at temporally localizing audible actions in videos. This task is crucial for automating the dubbing of actions, such as jumping and collisions, in silent videos, and for facilitating the re-dubbing of sounds in video editing. Our method, $TA^2Net$, employs inflectional flow as a foundational prior, establishing a promising benchmark for this task.

services like YouTube and TikTok. In particular, movies and television dramas often require dubbing, traditionally a labor-intensive process where professionals meticulously identify precise moments for audio insertion. This process is especially demanding for audible actions, such as the sounds of punches in boxing or weapon clashes. We argue that computers can autonomously mark these audible actions, significantly enhancing the efficiency of video clip dubbing, diverging from traditional manual line dubbing, as shown in Fig. 1.

Techniques specifically designed for video clip dubbing applications, particularly for detecting the precise temporal location of audible actions within a video, remain largely unexplored. The closest related fields within computer vision are video action recognition and temporal action localization. Current methods in action recognition (Girdhar et al., 2022; Huang et al., 2017; Lin et al., 2019a; Alwassel et al.,

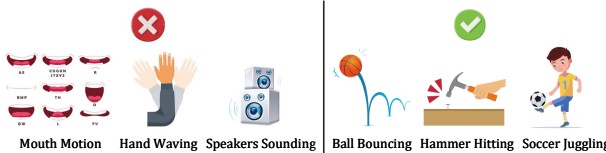

Figure 2: Instead of invisible audible actions (left), our study targets 'visible audible actions' (right part), where the correlation between visual movements and their associated sounds enables resolution through computer vision techniques. our research focuses on the computationally tractable challenges in audible actions.

2020; Feichtenhofer, 2020; Xu et al., 2020b; Bertasius et al., 2021; Ryali et al., 2023) excel at identifying different types of actions within a video. However, they primarily focus on the semantic aspects of actions rather than their specific kinematic characteristics. This emphasis leads to an oversight of the internal variations within actions, which are crucial for identifying and differentiating audible moments. On the other hand, video temporal action localization methods (Lin et al., 2018; 2019b; Xu et al., 2020a; Lin et al., 2021; Xia et al., 2022; Zhang et al., 2022; Cheng & Bertasius, 2022; Shao et al., 2023) are effective at discerning the temporal boundaries between actions, but they often fail to capture the essential spatiotemporal dynamics of specific motion frames that are critical for detecting audible actions. Audible actions are fine-grained moments within the entire action process, making them more challenging to localize compared to coarse-grained action boundaries.

In this paper, we present the new task of audible action temporal localization, aimed at predicting the frame-level positions of visible audible actions (see Fig. 2 for clearer definition). Central to our approach is the principle that visible audible actions, especially those involving collision sounds, are characterized by inflectional movements, such as abrupt changes during collisions. In response, we design $TA^2Net$, which explores kinematic analysis to develop an inflectional flow estimation method, leveraging the second derivative of displacement to detect changes in motion indicative of collisions, thereby enabling the model to identify sound positions without audio input. Additionally, we incorporate a self-supervised spatial localization strategy into our training process, enhancing the model's temporal representations through spatial domain analysis. This strategy employs spatial weight information to guide contrastive learning across both inter-video and intra-video contexts, with the side-output indicating the location of audible actions. To overcome the absence of precise labels for audible actions in existing datasets, we created a dataset called $Audible623$, comprising 623 videos with collision sound actions from Kinetics (Kay et al., 2017), UCF101 (Soomro et al., 2012), and YouTube, with audible actions marked at the frame

level and an average of 250 frames per video. We have extensively compared our method against existing techniques using our $Audible623$ dataset, where it demonstrated superior performance. Moreover, leveraging inflectional flow as an early indicator of motion change has proven essential for identifying periodic movements. Our model's effectiveness extends beyond the proposed task, as evidenced by its performance in traditional tasks such as repetitive counting in the UCFRep (Zhang et al., 2020) and CountixAV (Zhang et al., 2021) datasets. These results highlight the robustness and versatility of our framework.

In summary, our main contributions are fourfold:

- We present a new task of the audible action temporal localization and establish the first dedicated dataset, $Audible623$, specifically designed for this purpose.
- We tailor an inflectional flow estimation method grounded in the second derivative of the position-time image, aimed at enriching kinematic data with details on object state changes to mimic audible actions.
- We propose a novel auxiliary training method featuring self-supervised spatial localization, which utilizes acquired spatial information to boost the network's representational skills, additionally identifying collision spatial locations as a secondary output.
- We showcase the method's superior ability to temporally localize audible actions and its extensive applicability to other tasks, such as conventional repetitive action counting and sound source localization.

## 2. Related Work

**Temporal Action Analysis** is a crucial area in video understanding, focusing on the recognition, localization, and analysis of actions over time. Temporal action localization aims to identify the categories and localize the timing boundaries of action instances. Methods (Xu et al., 2017; Chao et al., 2018; Lin et al., 2018; 2019b; Xu et al., 2020a; Lin et al., 2021; Xia et al., 2022) first generate potential action proposals and then use a classifier to predict the action category. Other methods (Lin et al., 2017a; Liu & Wang, 2020; Tirupattur et al., 2021; Zhang et al., 2022; Cheng & Bertasius, 2022; Shao et al., 2023; Huang et al., 2024) achieve localization directly through an end-to-end manner. Zhang *et al.* (Zhang et al., 2022) propose a transformer-based method to classify moments in videos. Shao *et al.* (Shao et al., 2023) measure the action sensitivity to address the discrepant information of each frame. These methods focus on localizing the entire process of an action. In contrast, our approach focuses on identifying the precise timing point of audible actions within the action process. To be specific, the difference between the two tasks is that one focuses on event-level localization, while the other concentrates on accurate key frame identification. Action Counting is another aspect of

temporal action analysis, which quantify the frequency of specific actions in videos. Existing methods (Karvounas et al., 2019; Hu et al., 2022; Dwibedi et al., 2020; Zhang et al., 2020; 2021) estimate action counts by analyzing the periodicity and period lengths of actions. Hu *et al.* (Hu et al., 2022) suggest applying a transformer to encode multi-scale temporal correlation and further utilize a density map as representation to learn action period. Although the periodicity of actions is effective for analyzing the action process, it falls short in distinguishing the frame-level differences within actions.

**Sound Source Localization** (Zhao et al., 2018; Qian et al., 2020; Chen et al., 2021; Fedorishin et al., 2023; Oya et al., 2020; Lin et al., 2023) aims to identify the spatial location of the sound source in a video by analyzing the audio signal. Qian *et al.* (Qian et al., 2020) propose a two-stage audio-visual learning framework that performs cross modal feature alignment in a rough to fine manner. Chen *et al.* (Chen et al., 2021) use an automatic background mining technique with differentiable thresholds to incorporate regions with low correlation with a given sound into a negative set for comparative learning. Fedorishin *et al.* (Fedorishin et al., 2023) model the optical flow in videos as a prior to better assist in locating sound sources, and utilized cross attention to form stronger audio-visual correlations to achieve visual sound source localization. Here we propose an auxiliary training method that leverages spatial information and produces a source localization map as a side-output.

## 3. Audible Actions Dataset

**Data Collection.** As none of the existing video action datasets match the requirement of the proposed audible action temporal localization task, we need to collect an audible actions video dataset for training and evaluating our method. Initially, we gather videos from YouTube and existing action video datasets, including Kinetics (Kay et al., 2017) and UCF101 (Soomro et al., 2012), which cover a wide range of categories. Despite these datasets contain numerous videos, many do not include audible actions, and some actions lack a clear timing definition. To address this, we filter the videos and select those containing audible actions, such as drumming, tennis, and hammer throwing. The collected video should include at least one collision event, as we focus on detecting the timing of visual collisions. Several examples from our dataset are illustrated in Fig. 3.

**Dataset Annotation.** To meet the requirement of frame-level detection accuracy, annotations on video frames are crucial. Some datasets, such as THUMOS14 (Idrees et al., 2017) and ActivityNet (Caba Heilbron et al., 2015), provide annotation for the start/end times of actions. However, simply considering the midpoint is unsuitable for non-linear actions, and other motion descriptors are unreliable, neces-

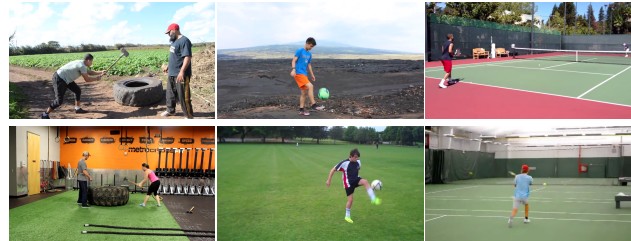

Figure 3: Examples from our $Audible623$ dataset. All videos include at least one visible audible action.

Table 1: Statistics comparison between our $Audible623$ and existing temporal action localization datasets (Idrees et al., 2017; Caba Heilbron et al., 2015) and repetitive counting dataset (Zhang et al., 2020).

|  | $Audible623$ | THUMOS14 | ActivityNet | UCFRep |
|---|---|---|---|---|
| # of Videos | 623 | 413 | 19,994 | 526 |
| # of Actions | 6,262 | 6,316 | 23,064 | 1,276 |
| Average Duration(s) | 9.2 | 4.3 | 49.2 | 8.2 |
| Audible Actions | ✓ | ✗ | ✗ | ✗ |
| Key Frame Labeling | ✓ | ✗ | ✗ | ✗ |

sitating human annotations. Since we do not use any audio for training and inference, human volunteers are invited to watch each video entirely and inspect the content frame by frame. Initially, volunteers are asked to determine whether the video contains at least one audible action. If a video couldn't be visually determined for the temporal location of audible frames, it is excluded. Volunteers are then tasked with labeling the frames for each audible action, and the annotations of all audible action frames are converted into keyframe labels, indicating whether a frame corresponds to an audible action.

**Dataset Statistics.** After collecting and annotating the action videos, we obtain a total of 623 videos and allocate 497 videos for training and 126 videos for evaluation. The videos in our dataset have durations ranging from 2.3 to 30.7 seconds, with an average duration of 9.2 seconds. On average, each video consists of 250 frames. Tab. 1 offers a statistical comparison between our dataset and commonly used datasets in temporal action localization. Our dataset furnishes precise keyframe annotations, a feature absent in existing datasets yet crucial for the task of audible action temporal localization.

## 4. Method

**Problem Formulation**. Given a silent video containing audible actions, our goal is to figure out the moment when audible action occurs. Furthermore, we propose a more challenging setting that aims to determine the moment at the frame level, i.e., whether each frame contains action that can generate sound. This contributes to achieving audio-

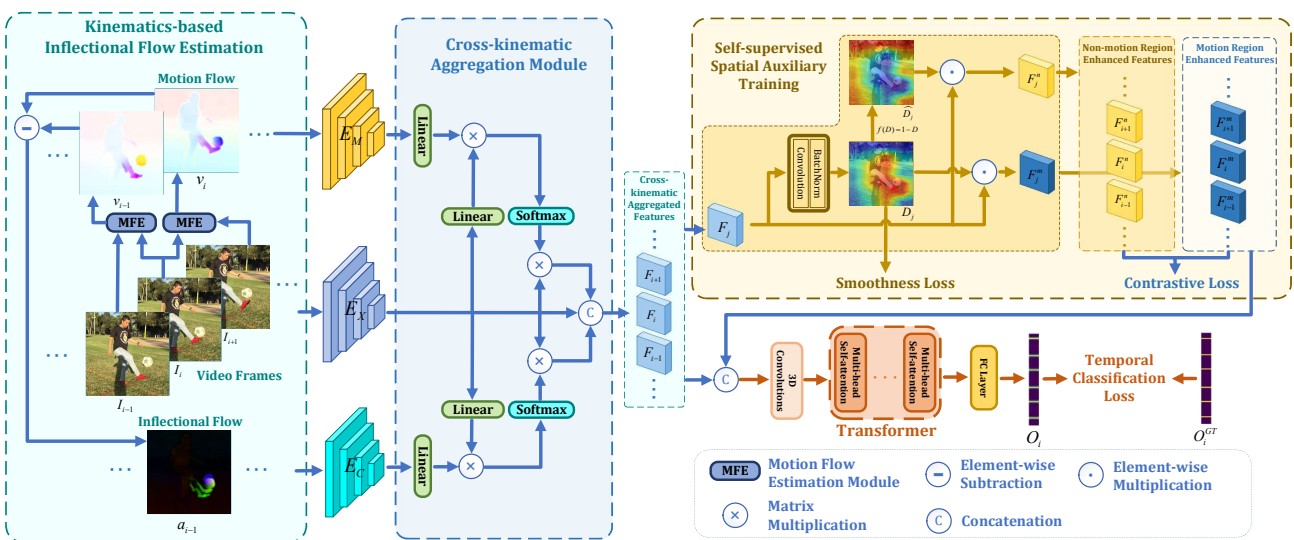

Figure 4: Overview of $TA^2Net$. Our model leverages the novel inflectional flow as the supplementary of kinematics prior (*e.g.*, motion) to identify the occurrence of audible actions. We first conduct the kinematics analysis to calculate the proposed inflectional flow as one of the kinematics priors under the sound-from-collision assumption. Then a cross-kinematics aggregation module is introduced to pop-up motion and velocity inflection information from the contexture features of the original image. To further model motion and velocity inflection in the spatial dimension, we propose a self-supervised spatial auxiliary training strategy to localize the motion of the object, both at inter-video level (contrastive loss) and intra-video level (smoothness loss). We then learn classification probabilities $O_i$ which indicate frames containing audible actions.

visual consistency. Concretely, given a video sequence $I \coloneqq \{I_t \in \mathbb{R}^{H \times W \times 3} \mid t = 1, 2, \ldots, T\}$ with length $T$, our task is to estimate probabilities for classifying frames into sound frames class and no-sound frames class, represented as $O \coloneqq \{O_t \in [0,1]^2 \mid t = 1, 2, \ldots, T\}$.

**Network Design**. The overview of our model is illustrated in Fig. 4. Our fundamental assumption posits that audible actions, specifically collisions, are typically induced by sudden changes in the forces acting upon an object, which manifest as inflections in velocity. Additionally, these collisions are the primary sources of sound. According to this, we delve into the velocity inflection and motion in both temporal and spatial prospective. For temporal modeling, we propose the concept of 'inflectional flow' to signify sudden changes in velocity of an object, enhancing traditional kinematic analysis by serving as a motion prior. This is complemented by the integration of backbone encoders and a cross-kinematic aggregation module, which together are designed to extract motion-guided contextual features. For spatial modeling, we introduce a self-supervised auxiliary training strategy, constraining the prediction of audible actions through both inter- and intra-video view.

### 4.1. Timing Audible Actions with Inflectional Flow

Based on our sound-from-collision assumption, detected sudden velocity changes can be important indicating factors

for identifying audible frames. Considering previous action recognition methods (Lin et al., 2019a; Feichtenhofer, 2020; Ryali et al., 2023) has not fully explored the potential inflection of velocity from motion, we propose to further complete traditional kinetic analysis (*i.e.*, only motion flow) by introducing the novel inflectional flow, motivated by (Xu et al., 2021).

**Inflectional Flow**. We consider the video as a sequence of the object's position over time, that is, a 2D-position-time graph (*i.e.*, $\mathbf{x}$-$t$ graph, where $\mathbf{x}$ is the 2D position vector). According to the laws of kinematics, we can get the corresponding velocity-time graph ($\mathbf{v}$-$t$ graph) by taking the first order derivative:

$$v(t) = \frac{d\mathbf{x}}{dt}. \tag{1}$$

Furthermore, the acceleration-time graph ($\mathbf{a}$-$t$ graph) is given by the second order derivative:

$$a(t) = \frac{d^2\mathbf{x}}{dt^2}. \tag{2}$$

According to Newton's First Law, the abrupt application of an external force, manifesting as a sudden change in acceleration, alters an object's state of motion. This concept underpins our analysis, as the acceleration function $a(t)$ yields critical insights into audible actions. Therefore, we characterize the velocity inflection point, represented by $a(t)$, as the 'inflectional flow'. We then integrate the

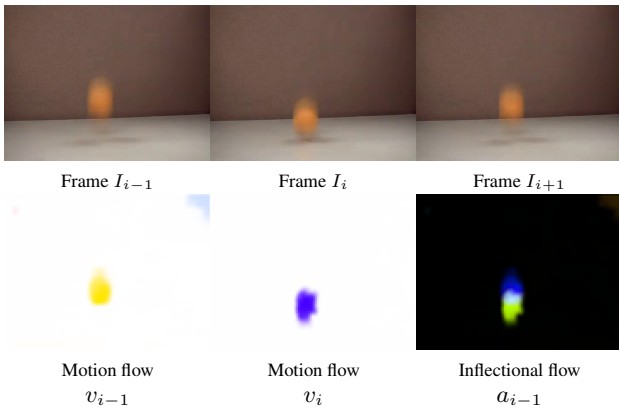

| Frame $I_{i-1}$ | Frame $I_i$ | Frame $I_{i+1}$ |

| Motion flow $v_{i-1}$ | Motion flow $v_i$ | Inflectional flow $a_{i-1}$ |

Figure 5: Visualization of the motion and inflectional flow. The kinematic analysis gives crucial action prior according to motion flow estimated by optical flow. The light part of $a_{i-1}$ illustrates the momentary change in the direction of motion.

inflectional flow, denoted as $a_t := a(t)$, with the motion flow, $v_t := v(t)$, serving as a novel kinematic prior for our model.

In practical, we utilize optical flow to measure $v(t)$ under the assumption that the time interval between two adjacent frames in a video is tiny enough. And we adopt the differential operation on $v(t)$ to obtain $a(t)$. Given three adjacent frames ($I_{i-1}$, $I_i$, $I_{i+1}$) of a video, we initially compute the forward optical flow (i.e., $I_{i-1}$ to $I_i$, and $I_i$ to $I_{i+1}$) as motion flow ($v_{i-1}^+$, $v_i^+$) using a pre-trained optical flow estimation network. Then the inflectional flow is given by:

$$a_{i-1}^+ = v_i^+ - v_{i-1}^+. \qquad (3)$$

We define the $v_{i-1}^+$ and $a_{i-1}^+$ as the kinematics prior. Similarly, we can also obtain the backward kinematics prior, including $v_i^-$, $v_{i+1}^-$, and $a_i^-$. The visualization of motion and inflectional flow is showed in Fig. 5.

We introduce three separate encoders ($E_X$, $E_M$, and $E_C$) with same structure to extract the context feature of the original frames, motion flow, and inflectional flow respectively. Each of these three encoders has the same structure, consisting of the first four layers of ResNet50 (He et al., 2016). Specifically, we feed the input tuple in forward direction ($I_i$, $v_i^+$, $a_i^+$) into the three encoders and obtain corresponding feature ($f_i$, $m_i^+$, $c_i^+$). And the motion and velocity inflection features in backward direction ($m_i^-$ and $c_i^-$) can also be calculated given input ($v_i^-$, $a_i^-$) similarly. We further construct the bidirectional motion feature $m_i$ and velocity inflection feature $c_i$ by the following concatenation operation:

$$m_i = \texttt{Concat}(m_i^-, m_i^+), c_i = \texttt{Concat}(c_i^-, c_i^+). \quad (4)$$

Note that we trim the input sequence at the beginning and the end to ensure that every $I_i$ has corresponding kinematics features (i.e., $m_i$ and $c_i$).

**Cross-Kinematics Aggregation**. To better leverage the guidance from the kinematics prior, we introduce the Cross-Kinematics Aggregation module, which aims to pop-up motion and velocity inflection information from image context feature. Specifically, given the tuple consists of image feature and kinematics features ($f_i$, $m_i$, $c_i$), we first project them into ($K_{f(i)}$, $Q_{m(i)}$, $Q_{c(i)}$) respectively using a linear layer. Then we calculate the relevance maps as:

$$A_{mf(i)} = \frac{Q_{m(i)}(K_{f(i)})^T}{\sqrt{d}}, A_{cf(i)} = \frac{Q_{c(i)}(K_{f(i)})^T}{\sqrt{d}}, \quad (5)$$

where $A_{mf(i)}$ and $A_{cf(i)}$ denote the attention matrix (Vaswani, 2017) for image-to-motion and image-to-inflection respectively. The spatial attention is formulated as:

$$\begin{aligned} h_{m(i)} &= V_{f(i)}\texttt{softmax}(A_{mf(i)}), \\ h_{c(i)} &= V_{f(i)}\texttt{softmax}(A_{cf(i)}), \end{aligned} \qquad (6)$$

where $h_{m(i)}$ and $h_{c(i)}$ is the cross-kinematics feature of image-to-motion and image-to-inflection respectively, and $V_{f(i)}$ is obtained by applying linear layer on $f_i$.

Finally, we concatenate the three features ($f_i$, $h_{m(i)}$, and $h_{c(i)}$) to the cross-kinematics feature, denoted as $F_i$.

### 4.2. Self-supervised Spatial Auxiliary Training

In addition to modeling velocity inflection through temporal dimension, we reconsider it in spatial dimension. Here, we adopt a self-supervised auxiliary training strategy, considering data labelling for the spatial location of audible action is time-consuming and difficult. Specifically, we enhance the discrimination ability of our model on the motion flow in the spatial dimension by conducting contrastive learning in motion and non-motion areas. On the other hand, we also observe that the motion state changes are continuous, thus a smoothing loss is introduced for constraining.

**Spatial Localization of Motion**. In previous temporal action localization methods (Zhang et al., 2022; Shao et al., 2023), they do not consider explicitly learning the spatial motion of the objects. Therefore, we suggest introducing spatial motion localization as an auxiliary training task which aims to reconstruct the motion of object. This strategy can effectively boost the network to distinguish the most discriminative area of object motion for better locating frames with potential audible actions.

Since reconstructing the precise motion of the object is challenging and unnecessary, instead of learning object segmentation mask, we relax the constraint by turning to learn a discriminative map, where the motion region is activated with higher probability. In detail, given $k$ cross-kinematics

aggregated features $\{F_j \mid j = 1, 2, \ldots, k\}$, the corresponding discriminative maps $\{D_j \mid j = 1, 2, \ldots, k\}$ through a $3 \times 3$ convolution layer, a batch normalization layer, and Sigmoid operation. Here, these $k$ frames are not necessarily from the same video.

**Inter-Video Contrastive Learning**. We further conduct the contrastive learning on the motion-area (high-probability region) and non-motion area (low-probability region) on the discriminative map. We first mask the cross-kinematics aggregated feature $F_j$ by both motion region and non-motion region given by $D_j$ separately, formulated as:

$$F_j^m = D_j \otimes F_j, F_j^n = (1 - D_j) \otimes F_j, \qquad (7)$$

where $\otimes$ is the Hadamard product. $F_j^m$ and $F_j^n$ denote the motion region activated and non-motion region activated aggregated features respectively. Given $F^m := \{F_p^m \mid p = 1, 2, \ldots, k\}$ and $F^n := \{F_q^n \mid q = 1, 2, \ldots, k\}$, the objective of negative contrast (motion region and non-motion region) is given by:

$$\mathcal{L}_{nc} = -\frac{1}{k^2} \sum_{p=1}^{k} \sum_{q=1}^{k} \log(1 - \langle F_p^m, F_q^n \rangle), \qquad (8)$$

where $\langle F_p^m, F_q^n \rangle = \frac{F_p^m \cdot F_q^n}{\|F_p^m\|\|F_q^n\|}$. Furthermore, the positive contrast objective is formulated as:

$$\mathcal{L}_{pc}(F) = -\frac{1}{k(k-1)} \sum_{p=1}^{k} \sum_{q=1}^{k} \mathbb{1}_{[p \neq q]} w_{p,q} \log(\langle F_p, F_q \rangle), \qquad (9)$$

where $\mathbb{1}_{[p \neq q]}$ is 1 for $p \neq q$ and otherwise 0, and $F$ can be either $F^m$ or $F^n$. $w_{p,q}$ represents weights to penalize positive region pairs with less cosine similarity given by ranking, formulated as:

$$w_{p,q} = \exp(-\alpha \cdot \text{rank}(\langle F_p, F_q \rangle)), \qquad (10)$$

where $\alpha$ is a hyper-parameter for smoothness controlling and $\text{rank}(\langle F_p, F_q \rangle)$ represents the rank (Xie et al., 2022) of $\langle F_p, F_q \rangle$ among all the cosine similarity pair of $F^m$ or $F^n$.

The above equations define the objectives for motion to motion region ($\mathcal{L}_{pc}^m := \mathcal{L}_{pc}(F^m)$) and non-motion to non-motion region ($\mathcal{L}_{pc}^n := \mathcal{L}_{pc}(F^n)$). Finally, the total contrastive loss $\mathcal{L}_{cont.}$ is:

$$\mathcal{L}_{cont.} = \mathcal{L}_{nc} + \mathcal{L}_{pc}^m + \mathcal{L}_{pc}^n. \qquad (11)$$

**Intra-Video Smoothing Constraint**. Notably, the actions in video are typically continuous. Therefore, we refine the localization results to ensure that there are no significant changes in the located regions between consecutive frames,

ensuring accurate localization. We further propose an intra-video temporal regularization loss:

$$\mathcal{L}_{temp.} = \sum_i \|D_{i+2} + D_i - 2D_{i+1}\|_2, \qquad (12)$$

where $i$ is the index of a frame from the same video.

The result is feed back to the audible action prediction part and train as a supervised way. Therefore, we can re-detect the action frame by separated action regions.

### 4.3. Spatio-Temporal Fusion

Upon obtaining the aggregated features $\{F_j \mid j = 1, 2, \ldots, k\}$ and motion region activated features $\{F_j^m \mid j = 1, 2, \ldots, k\}$, we utilize a 3D convolution to encode the temporal and spatial motion information. To extract more detailed information on audible actions, inspired by (Hu et al., 2022), we channel the outputs of 3D convolution into a transformer network equipped with self-attention modules. This setup facilitates the integration of spatiotemporal features, effectively pinpointing potential instances of audible actions within the video. Then we forward them through a fully connected layer to produce the final output $O_i$.

### 4.4. Objective Function

For the frame-wise prediction results in the temporal domain, we conduct supervised training using cross-entropy loss and focal loss (Lin et al., 2017b) as follows:

$$\mathcal{L}_{action} = \lambda_{ce}\mathcal{L}_{ce} + \lambda_{focal}\mathcal{L}_{focal}, \qquad (13)$$

where $\lambda_{ce}$ and $\lambda_{focal}$ are weighting parameters for loss terms, which are set to 1 and 0.1, respectively. It is noteworthy that we employ a soft label technique based on Gaussian augmentation to calculate the $\mathcal{L}_{ce}$.

In summary, the total loss for our $TA^2Net$ is a weighted sum of three losses:

$$\mathcal{L}_{total} = \lambda_{action}\mathcal{L}_{action} + \lambda_{cont.}\mathcal{L}_{cont.} + \lambda_{temp.}\mathcal{L}_{temp.}, \qquad (14)$$

where $\lambda_{action}$, $\lambda_{cont.}$ and $\lambda_{temp.}$ are weighting parameters for corresponding loss terms, which are set to 1, 0.01 and 0.002, respectively.

## 5. Experiment

**Implementation Details.** We encode kinematic information using a pre-trained ResNet50 (He et al., 2016), mapping it to 256 dimensions in the cross-kinematics module. During training, we randomly sample 64 frames per video, resizing them to $112 \times 112$ pixels. The model is trained using the Adam optimizer (Kingma & Ba, 2015) with a learning rate of 5e-6, a batch size of 4, and 20k iterations. For inference, we segment the video continuously with a step size of 64,

padding the final segment if it is shorter than 64 frames. For optical flow extraction, we employ GMFlow (Xu et al., 2023) due to solid empirical performance and a good trade-off between accuracy and efficiency. We implement our method on PyTorch with CUDA, version 1.13. All experiments were conducted on a single NVIDIA A800 GPU with 80GB of memory, under Ubuntu 20.04 as the operating system. For model training, we trained the proposed method for 200 epochs, which took approximately 4 hours.

**Metrics**. We introduce five metrics to evaluate the performance of audible action temporal localization: Recall, Precision, $F_1$, Number Match Error (NME), and Position Match Error (PME). Recall measures the method's ability to detect all audible action frames, while Precision evaluates the proportion of accurate predictions among all predicted frames. The $F_1$ score balances precision and recall to provide an overall performance measure. For quantitative and temporal accuracy, NME quantifies the difference between predicted and ground truth audible action frame counts, and PME assesses temporal accuracy, allowing a detection time gap of up to 2 frames from the ground truth. Additionally, for counting evaluation, we use mean absolute error (MAE) and off-by-one accuracy (OBO) following the approach in (Dwibedi et al., 2020; Hu et al., 2022). MAE measures the absolute discrepancy between predicted and ground truth counts, normalized by the ground truth, while OBO indicates the proportion of correctly counted instances within one count of the ground truth. Further details are provided in the **appendix**.

### 5.1. Comparisons with Existing Methods

In this section, we employ several methods for both qualitative and quantitative comparison with our proposed approach. To the best of our knowledge, there is currently no work directly aligning with our setting. Hence, we select eleven methods of four types for comparison based on their relevance to the audible action temporal localization task, including temporal action localization (BMN (Lin et al., 2019b), ActionFormer (Zhang et al., 2022) and TriDet (Shi et al., 2023)), repetitive action counting (RepNet (Dwibedi et al., 2020) and TransRAC (Hu et al., 2022)), video/action recognition (TimeSformer (Girdhar et al., 2022), X3D (Lin et al., 2019a), Omnivore (Feichtenhofer, 2020), TSM (Bertasius et al., 2021) and Hiera (Ryali et al., 2023)), and anomaly detection (STG-NF (Hirschorn & Avidan, 2023)). For a fair comparison, we load the corresponding pretrained model (if available) and fine-tuned for the same epochs on $Audible623$ dataset. All compared methods are trained with the same supervision using our key frame annotations. More details can be found in the **appendix**.

**Quantitative Comparison**. We first compare our method with above eleven methods quantitatively. As shown in

Table 2: Quantitative comparison with associated methods on $Audible623$ dataset.

| Methods | Recall↑ | Precision↑ | $F_1$ ↑ | NME↓ | PME↓ |
|---|---|---|---|---|---|
| BMN | 0.417 | 0.486 | 0.413 | 9.327 | 0.951 |
| ActionFormer | 0.323 | 0.357 | 0.294 | 14.663 | 1.162 |
| TriDet | 0.459 | 0.338 | 0.328 | 11.378 | 1.306 |
| RepNet | 0.377 | 0.428 | 0.362 | 10.319 | 1.100 |
| TransRAC | 0.569 | 0.614 | 0.553 | 12.176 | 0.988 |
| Omnivore | 0.410 | 0.368 | 0.330 | 19.454 | 1.256 |
| TSM | 0.314 | 0.372 | 0.303 | 12.160 | 1.153 |
| X3D | 0.426 | 0.435 | 0.392 | 10.370 | 1.039 |
| TimeSformer | 0.470 | 0.455 | 0.405 | 16.101 | 1.172 |
| Hiera | 0.562 | 0.436 | 0.424 | 20.748 | 1.265 |
| STG-NF | **0.679** | 0.229 | 0.342 | - | - |
| **Ours** | 0.648 | **0.656** | **0.616** | **3.462** | **0.744** |

Table 3: Counting performance on UCFRep and CountixAV datasets. Top-2 results are marked in **bold** and underlined. Our method has not been trained on any counting dataset.

| Method | UCFRep | | CountixAV | |
|---|---|---|---|---|
| | MAE↓ | OBO↑ | MAE↓ | OBO↑ |
| RepNet | 0.915 | 0.074 | 0.749 | 0.231 |
| TransRAC | 0.594 | **0.222** | 0.686 | 0.255 |
| X3d | 1.245 | 0.037 | 0.876 | 0.192 |
| TimeSformer | 0.832 | 0.0 | 1.551 | 0.185 |
| Hiera | 0.791 | 0.152 | 3.648 | 0.231 |
| **Ours** | **0.588** | 0.185 | **0.549** | **0.346** |

Tab. 2, the related methods of temporal action localization (Lin et al., 2019b; Zhang et al., 2022; Shi et al., 2023) perform bad quality, as they are designed for action recognition and localization in long videos. Repetitive action counting methods (Dwibedi et al., 2020; Hu et al., 2022) perform well in classification metrics. However, they excel in predicting the information of a single action sequence rather than the instantaneous action of collision. Consequently, they can only recognize the approximate position of the action and cannot accurately locate it, leading to suboptimal performance in NME. The metrics for video recognition methods, especially those represented by Hiera (Ryali et al., 2023) are low. This can be attributed to the limitation of these methods to the utilization of long time sequence action content and their lack of sensitivity to frame-level features. Concerning the video anomaly detection method STG-NF (Hirschorn & Avidan, 2023), while the human keypoint sequence effectively represents actions, it struggles to discern action changes at the moment of collision. Notably, we do not report NME and PME metrics, as they are unfair for snippet level detection. Thanks to the inflectional flow estimation method and the self-supervised spatial auxiliary training strategy proposed in our approach, our method outperforms existing methods in four key metrics, demonstrating its superiority. Our network effectively utilizes temporal kinematic information in action videos,

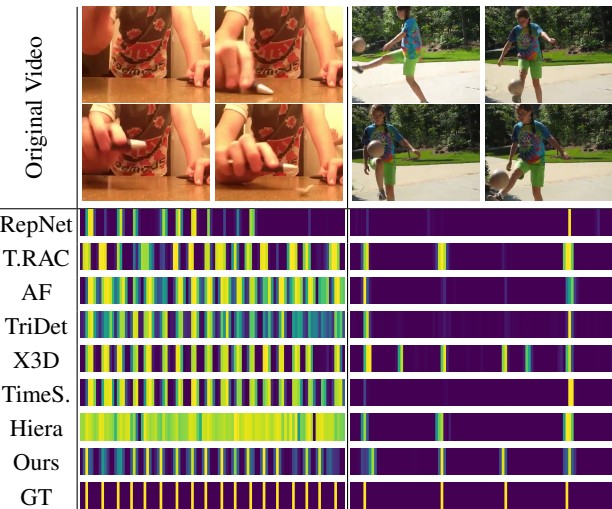

Figure 6: Qualitative comparison of the visualization of temporal predictions on $Audible623$ dataset. The first row shows several frames of the input videos, and the second row shows the visualization of temporal probabilities ($O_i$).

Table 4: Ablation results on $TA^2Net$.

| Video | Motion | Infle. | Aggr. | $\mathcal{L}_{cont.}$ | $\mathcal{L}_{temp.}$ | Recall↑ | Prec.↑ | $F_1$ ↑ | NME↓ | PME↓ |
|---|---|---|---|---|---|---|---|---|---|---|
| ✓ | | | | | | 0.496 | 0.395 | 0.406 | 5.992 | 1.021 |
| ✓ | ✓ | | | | | 0.600 | 0.500 | 0.501 | 5.748 | 0.898 |
| ✓ | ✓ | ✓ | | | | 0.609 | 0.578 | 0.561 | 4.420 | 0.781 |
| ✓ | ✓ | ✓ | ✓ | | | 0.640 | 0.597 | 0.587 | 3.773 | 0.806 |
| ✓ | ✓ | ✓ | ✓ | ✓ | | 0.638 | 0.632 | 0.601 | 3.731 | 0.756 |
| ✓ | ✓ | ✓ | ✓ | ✓ | ✓ | **0.648** | **0.656** | **0.616** | **3.462** | **0.744** |

tify the approximate range of actions. This limitation arises because their designs focus solely on the duration of actions, resulting in the identification of adjacent frames as audible actions as well. Unlike all the competitors, our method achieves the highest accuracy, which is attributed to the spatial and temporal guidance of kinematics prior obtained through inflectional flow estimation and self-supervised spatial auxiliary training. And we have a fine-grained timing prediction for all the audible actions. More comparisons can be found in **appendix**.

### 5.2. Ablation Studies

**Inflectional Flow Estimation Module**. To evaluate the impact of the motion and inflectional flow estimation components, we tested a scenario where prior motion (Motion) and velocity inflection (Infle.) information were excluded, and then progressively integrated this kinematic data. As shown in the first three rows of Tab. 4, the baseline model, which relies only on appearance and temporal information from the video frames, performs the worst among all ablated models. Incorporating motion and velocity inflection data separately improves classification performance by better capturing motion changes. Notably, combining both motion and velocity inflection yields the best results. In continuous motion videos, where adjacent frames often change minimally, this component helps focus on frame-to-frame discrepancies, effectively capturing key actions in audible action sequences.

**Cross-kinematics Aggregation Module**. We also evaluated the cross-kinematics aggregation module (Aggr.). Here, we directly connected the visual and kinematic features extracted by the encoder to predict sound moments. As shown in the fourth row of Tab. 4, this module enhances performance by focusing on key content features through motion and velocity inflection, effectively capturing audible behaviors in the video.

**Auxiliary Training Strategy with Localization**. To assess the effectiveness of our auxiliary training strategy, we analyzed the contributions of inter-video contrastive loss ($\mathcal{L}_{cont.}$) and intra-video temporal regularization loss ($\mathcal{L}_{temp.}$). As shown in the last two rows of Tab. 4, both strategies significantly improve prediction accuracy, particularly precision. Inter-video contrastive learning enhances

combining it with spatial motion information to achieve superior results.

**Generalization to Counting Task.** We demonstrate that the proposed inflectional flow can effectively generalize to repetitive counting tasks. Without any fine-tuning, we directly compared our method to established counting techniques, including RepNet (Dwibedi et al., 2020) and TransRAC (Hu et al., 2022), on the audible action part of repetitive counting datasets. As shown in Tab. 3, our approach set new benchmarks, ranking (1st, 2nd) on UCFRep and (1st, 1st) on CountixAV, despite not being trained for repetitive counting. It's important to highlight that RepNet and TransRAC are specifically designed and trained for repetitive counting tasks. The success of our model's zero-shot predictions underscores the effectiveness of modeling inflectional flows at both temporal and spatial levels. This capability allows for precise identification of keyframes in repetitive actions, such as pinpointing the exact moment of collision in repeated tapping actions, which is crucial for accurate counting.

**Qualitative Comparison**. We also conduct a qualitative comparison with seven methods, including RepNet (Dwibedi et al., 2020), TransRAC (Hu et al., 2022), ActionFormer (Zhang et al., 2022), TriDet (Shi et al., 2023), TimeSformer (Bertasius et al., 2021), X3D (Feichtenhofer, 2020) and Hiera (Ryali et al., 2023). The visualization of the comparative results of temporal localization is shown in Fig. 6. We observe that Hiera fails in the first case, while RepNet mispredicts many actions. ActionFormer and TriDet tend to over-detect audible frames. Although X3D and TransRAC exhibit good prediction accuracy, they can only iden-

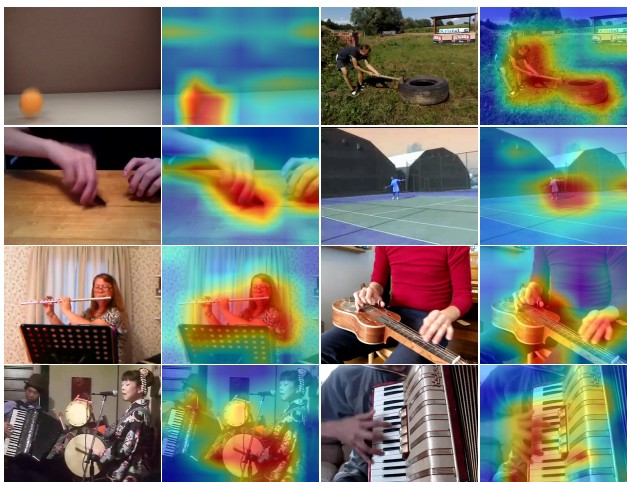

Figure 7: Visualization of spatial localization results on $Audible623$ (row 1-2) and AVE (Tian et al., 2018) (row 3-4) datasets. Our method shows effective spatial localization on the AVE dataset without specific training.

the detection of motion state changes by emphasizing spatial positional information, while intra-video constraints help smooth spatial localization, maintaining motion continuity.

### 5.3. Applications

In this section, we demonstrate the versatility of our method in various applications, including spatial localization and non-audible action analysis.

**Motion Spatial Localization.** Our proposed auxiliary training strategy effectively discriminates movement areas, as shown in Fig. 7. The spatial localization maps, produced as a side output of our model on the $Audible623$ and AVE datasets (Tian et al., 2018), demonstrate the effectiveness of this strategy in inferring the spatial position of motion within videos containing audible actions. Notably, our method was not trained on the AVE dataset, which highlights the excellent generalization ability of the proposed approach.

**Beyond Audible Actions.** Moreover, we highlight that inflectional flow extends beyond audible actions. As shown in Fig. 8, it performs robustly in non-audible actions as well, demonstrating its versatility across various types of motion analysis. Inflectional flow provides a detailed perspective on state changes within actions, capturing subtle transitions and variations throughout sequences, making it a valuable tool for comprehensive temporal action analysis.

## 6. Conclusion

In this paper, we introduce the task of audible action temporal localization to solve the problem of dubbing silent videos with visible actions and further propose its dedicated dataset

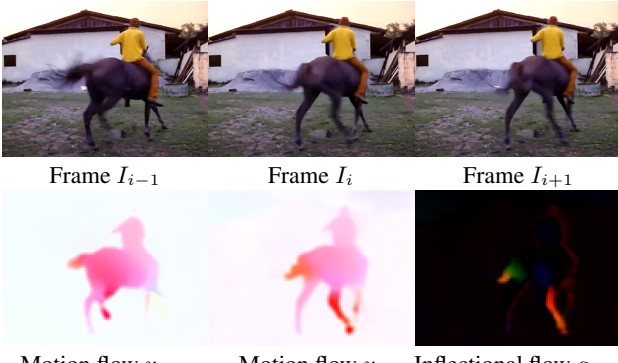

| Frame $I_{i-1}$ | Frame $I_i$ | Frame $I_{i+1}$ |
|---|---|---|
| Motion flow $v_{i-1}$ | Motion flow $v_i$ | Inflectional flow $a_{i-1}$ |

Figure 8: Inflectional flow for non-audible actions. By modeling visual motion dynamics, it can also captures subtle motion transitions across frames and flow fields, enabling robust temporal analysis.

($Audible623$) and a strong baseline method ($TA^2Net$). Our $TA^2Net$ utilizes kinematic analysis for inflectional flow estimation, identifying potential sound-producing events. We further enhance this with a self-supervised spatial localization strategy, applying motion analysis at both inter- and intra-video levels. Experimental results demonstrate our model's excellence in temporal localization of audible actions and its adaptability to other tasks, such as repetitive action counting and sound source localization.

**Limitation.** Our approach leverages optical flow to estimate the motion flow for further calculating inflectional flow. Therefore, the accuracy of optical flow prediction will affect the prediction quality of the final model. For those challenging video cases like large viewing angle changes, the inaccurate optical estimation will further affect the prediction quality of our model. In the future, we will explore more robust motion flow estimation methods.

## Impact Statement

This paper presents work whose goal is to advance the field of Machine Learning. There are many potential societal consequences of our work, none which we feel must be specifically highlighted here.

## Acknowledgement

This work is supported by the National Natural Science Foundation of China (Grant 62472180), Guangdong Basic and Applied Basic Research Foundation (Grant 2024A1515011995), Guangdong Natural Science Funds for Distinguished Young Scholars (Grant 2023B1515020097), the National Research Foundation, Singapore under its AI Singapore Programme (AISG Award No: AISG3-GV-2023-011), and the Lee Kong Chian Fellowships.

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

## A. More details of dataset

We provide the action categories of videos in the proposed $Audible623$ dataset in Fig. 9. It should be noted that although $Audible623$ provides action category annotations, the category information is not used during training or evaluation. The $Audible623$ dataset comprises 14 categories of audible action videos and 64% of videos have a duration over 10 seconds.

We use two datasets, UCFRep (Zhang et al., 2020) and CountixAV (Zhang et al., 2021), to conduct experiments on repetitive counting tasks. The UCFRep (Zhang et al., 2020) dataset comprises 526 action videos from 23 categories sourced from the UCF101 (Soomro et al., 2012) dataset, each video is annotated with the intervals of repetitive actions. The CountixAV (Zhang et al., 2021) dataset consists of 1,863 repetitive action videos (some videos have become unavailable) from the Countix (Dwibedi et al., 2020) dataset by filtering out videos with unclear sound or background noise. Then the videos are labeled with the count of repetitive actions. Note that we use these videos with sounds removed as input.

Furthermore, we apply AVE (Tian et al., 2018) dataset for the visualization of spatial localization results. AVE (Tian et al., 2018) comprises 4,143 videos of visual events and the corresponding audio, covering 28 action categories. Each category includes at least 60 videos.

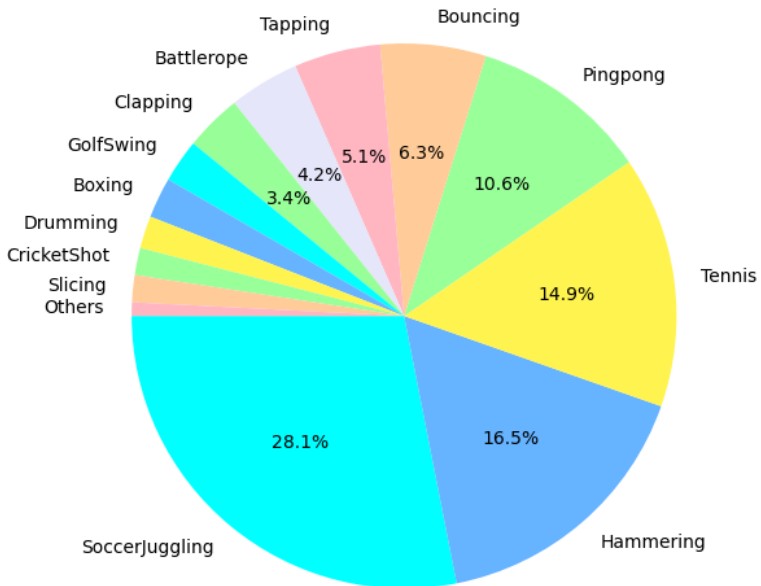

Figure 9: The distribution of videos in different categories of $Audible623$ dataset.

## B. Metrics

We utilize five metrics to evaluate our proposed Audible Action Temporal Localization task, including Recall, Precision, $F_1$, Number Match Error (NME), and Position Match Error (PME). These metrics are formulated as follows:

$$NME = \frac{1}{N} \sum_{i=1}^{N} |c_i - c_i^{GT}|, \tag{15}$$

$$PME = \frac{1}{N} \sum_{i=1}^{N} (\frac{1}{m_i} \sum_{j=1}^{m_i} |l_j - l_j^{GT}|), \tag{16}$$

where $N$ and $m_i$ are the number of videos and matched audible action frames, respectively. $c_i = \sum_{t=1}^{T_i} O_t$ and $c_i^{GT} = \sum_{t=1}^{T_i} O_t^{GT}$ indicate predictions and ground truths (GTs) of the sum of audible actions, respectively. $l_j$ and $l_j^{GT}$ are predictions and GTs of the temporal location of audible action frames, respectively.

Table 5: Ablation study over the weight of $\mathcal{L}_{cont.}$ and $\mathcal{L}_{temp.}$.

| $\lambda_{cont.}$ | $\lambda_{temp.}$ | Recall↑ | Prec.↑ | $F_1$ ↑ | NME↓ | PME↓ |
|---|---|---|---|---|---|---|
| 0 | 0 | 0.640 | 0.597 | 0.587 | 3.773 | 0.806 |
| 0.1 | 0 | 0.628 | 0.629 | 0.591 | 3.655 | 0.762 |
| 0.01 | 0 | 0.638 | 0.632 | 0.601 | 3.731 | 0.756 |
| 0.001 | 0 | 0.639 | 0.614 | 0.592 | 4.168 | 0.756 |
| 0.01 | 0.02 | 0.636 | 0.628 | 0.596 | 3.932 | 0.766 |
| 0.01 | 0.0002 | **0.671** | 0.620 | 0.607 | 3.924 | 0.792 |
| 0 | 0.002 | 0.625 | 0.627 | 0.587 | 4.252 | 0.772 |
| 0.1 | 0.002 | 0.638 | 0.617 | 0.597 | 3.487 | 0.754 |
| 0.001 | 0.002 | 0.667 | 0.611 | 0.603 | 4.622 | 0.820 |
| 0.01 | 0.002 | 0.648 | **0.656** | **0.616** | **3.462** | **0.744** |

Following the previous repetitive counting work (Dwibedi et al., 2020; Hu et al., 2022), the Off-By-One Error (OBO) and Mean Absolute Error (MAE) metrics for action counting can be also formulated as:

$$OBO = \frac{1}{N} \sum_{i=1}^{N} \left[ \left| c_i - c_i^{GT} \right| \leq 1 \right], \tag{17}$$

$$MAE = \frac{1}{N} \sum_{i=1}^{N} \frac{\left| c_i - c_i^{GT} \right|}{c_i^{GT}}. \tag{18}$$

## C. Comparison Methods

The relevant details for all comparison methods are as follows:

**Temporal Action Localization.** We choose three temporal action localization methods, including BMN (Lin et al., 2019b), ActionFormer(Zhang et al., 2022) and TriDet (Shi et al., 2023). Specifically, we employ them to frame-level features input and predict per frame performance. Specifically, for BMN, we use a two-stream network that incorporates both video and optical flow features as input.

**Repetitive Action Counting.** We select two state-of-the-art repetitive counting methods for comparison, including RepNet (Dwibedi et al., 2020) and TransRAC (Hu et al., 2022). These methods predict the number of repeated actions in the video by aggregating the probabilities of the final output. We adjust the final classifier to ensure it outputs frames with probabilities greater than a predefined threshold.

**Video/Action Recognition.** We adopt five state-of-the-art video/action recognition methods, including TimeSformer (Bertasius et al., 2021), X3D (Feichtenhofer, 2020), Omnivore (Girdhar et al., 2022), TSM (Lin et al., 2019a) and Hiera (Ryali et al., 2023), which recognize the categories of actions present in the video, such as shaking hands, hugging, etc. Similarly, we keep the original feature extraction module unchanged and only adapt the final classifier to make frame-level predictions.

**Anomaly Detection.** We choose STG-NF (Hirschorn & Avidan, 2023), a video anomaly detection method that solely utilizes human pose (Sun et al., 2024) to learn spatial and temporal relationships. Specifically, we extract human pose keypoints for videos in the $Audible623$ dataset and modify the sliding window to detect the presence of audible actions within the window.

## D. More Ablation Study

We further report the results of the ablation study over the weight of inter-video contrastive loss ($\mathcal{L}_{cont.}$) and intra-video temporal regularization loss ($\mathcal{L}_{temp.}$) in Tab. 5. The results demonstrate that adding $\mathcal{L}_{cont.}$ and $\mathcal{L}_{temp.}$ can improve the performance of our method. However, using only $\mathcal{L}_{temp.}$ resulted in increased recall but decreased precision, suggesting that solely maintaining spatial attention unchanged might lead to inaccurate localization.

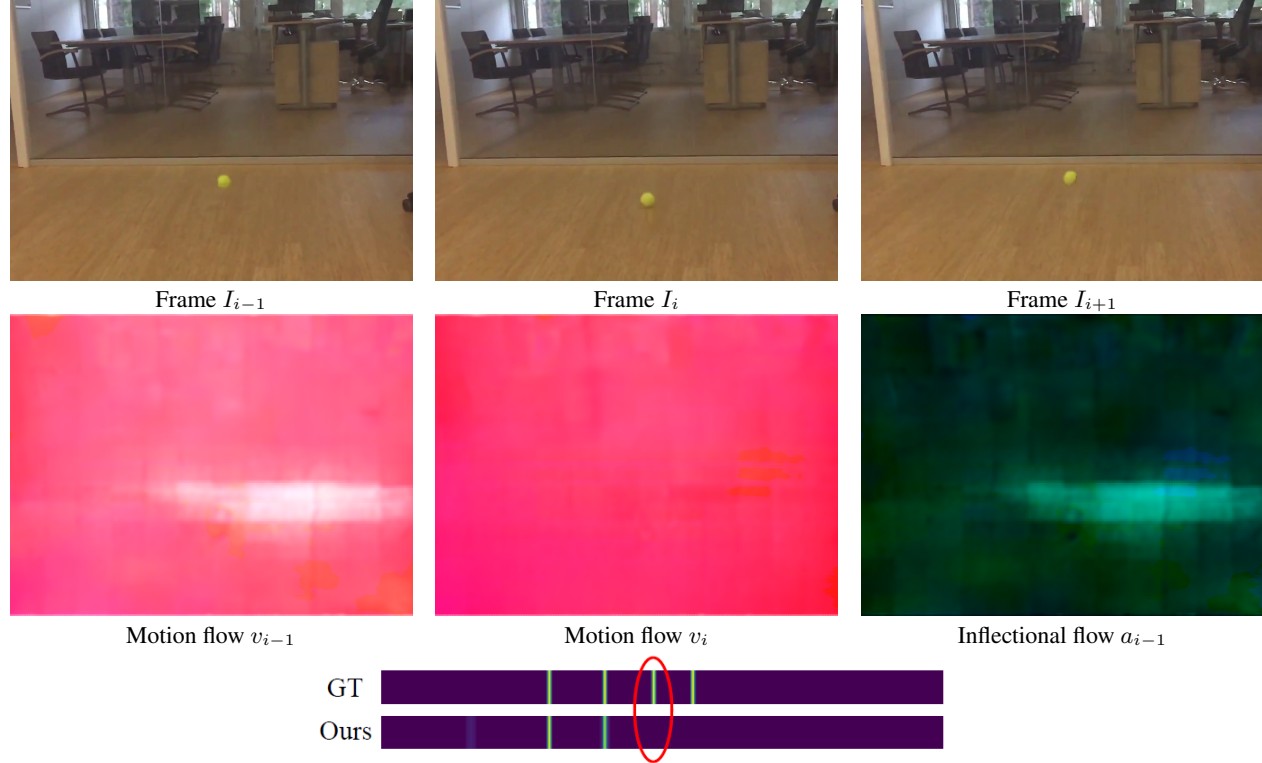

Figure 10: Visualization of failure case.

## E. More Qualitative Comparison

We also present more qualitative comparisons, visualizing temporal predictions on the proposed $Audible623$ dataset. We provide video frames and temporal prediction results, conducting a comparison between our method and several existing approaches (Zhang et al., 2022; Shi et al., 2023; Dwibedi et al., 2020; Hu et al., 2022; Feichtenhofer, 2020; Bertasius et al., 2021; Ryali et al., 2023; Zheng et al., 2023). These comparisons are illustrated in Fig. 12 and Fig. 13. It is evident from the visualizations that our method outperforms these approaches.

## F. Failure Case

As our method is based on the kinematic analysis for inflectional flow estimation, it relies on the accuracy of inflectional flow. We illustrate a failure instance in Fig. 10. The perceptible decline in the quality of optical flow estimation stems from camera viewpoint adjustments. Consequently, this degradation significantly impacts our kinematic analysis, resulting in our method's failure to detect audible actions in frame $I_i$ (highlighted by a red circle).

## G. Evaluation using Vision Language Model

We also test the performance of a large-scale video understanding model (Video-LLaMA (Zhang et al., 2023)) on the task of audible action temporal localization. As shown in Fig. 11, despite the special requirements (localizing the exact time frame of audible actions) made in the prompt, the results indicate that while these large models can broadly describe the actions occurring in a video, they lack the capability to accurately pinpoint the timing of actions, especially the precise moments of audible action.

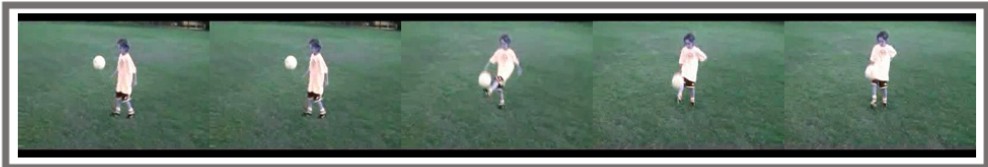

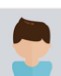

Based on the video content provided, please analyze the video to pinpoint the point in time of each kicking action, output the exact frame number where the ball was kicked in each kicking action, and then compile this information in the order it occurred.

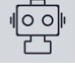

The following is a compilation of the exact frame numbers where the ball was kicked in each kicking action in the provided video, in the order it occurred:
Video Content: A young boy is shown playing soccer on a green grass field. He is wearing a white shirt and shorts. He is kicking the ball and running around on the field.
Kicking Action1: Frame 20-22 Kicking Action2: Frame 30-32 Kicking Action 3: Frame 40-42
Kicking Action4: Frame 50-52 Kicking Action5: Frame 60-62 Kicking Action 6: Frame 70-72

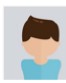

Actually, the exact frame numbers in the video where the ball is kicked in each action are 11, 28, 44, 61, 79.

Figure 11: Applying Video-LLaMA (Zhang et al., 2023) to Temporal Audible Action Localization Task. We found it failed to accurately localize the exact time frame of audible actions.

## H. Demo

We present the visualization of audible action temporal predictions and spatial localization in dynamic form in the supplementary video. Additionally, we demonstrate some results on video re-dubbing as a key application to fully exploit our model.

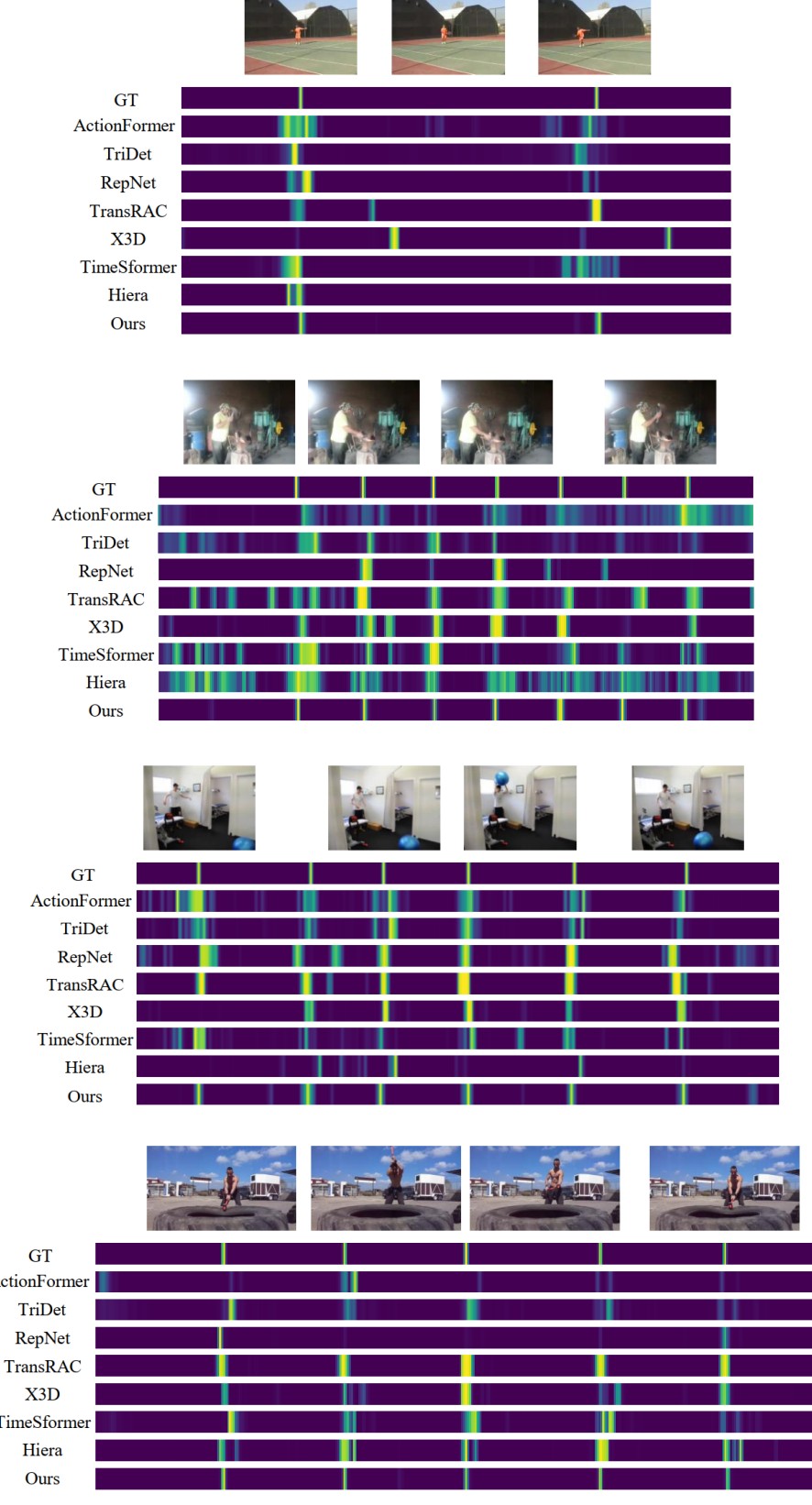

Figure 12: Visualization of temporal predictions on the $Audible623$ dataset.

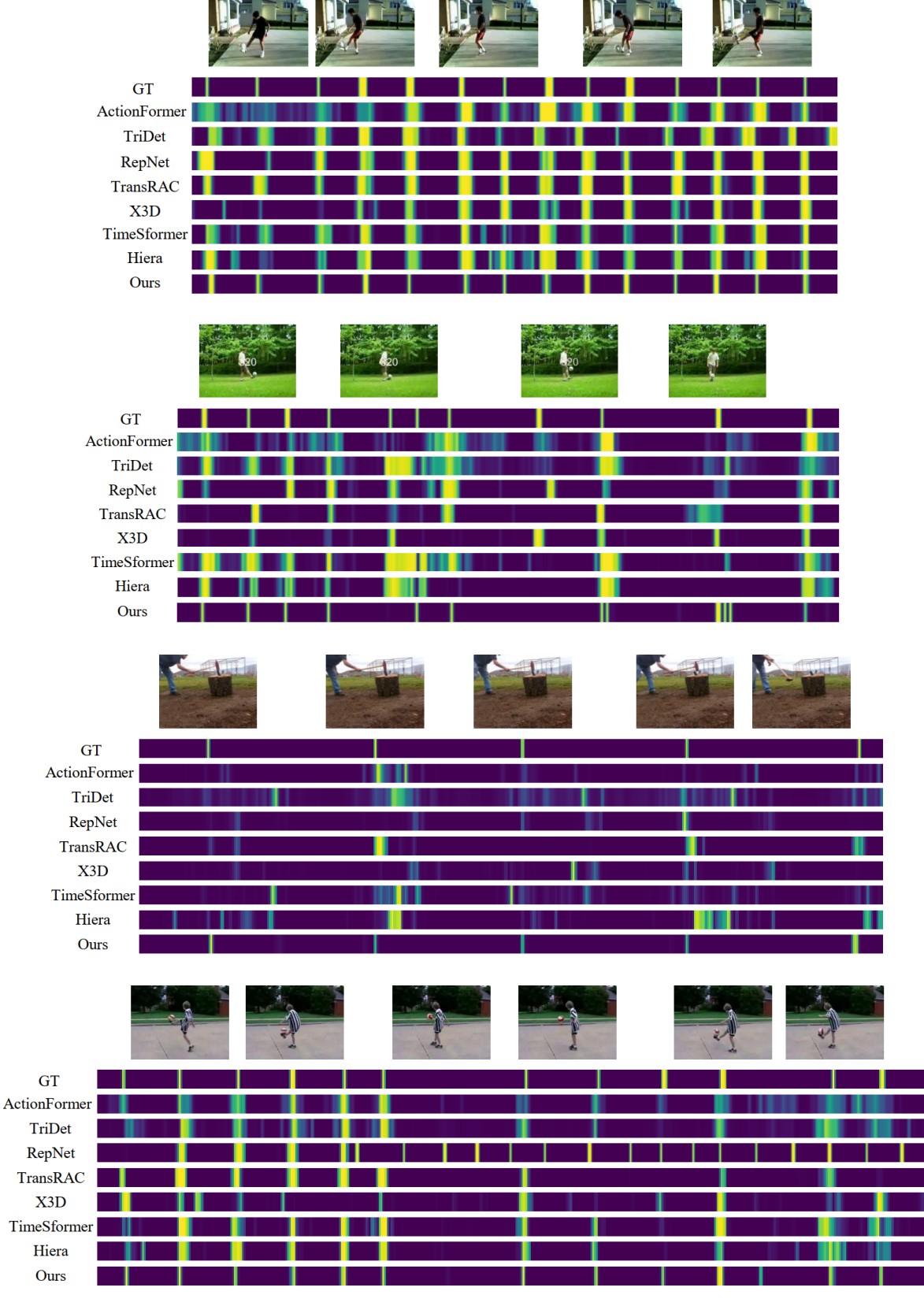

Figure 13: Visualization of temporal predictions on the $Audible623$ dataset.

