# OpenReview forum: "Action Dubber: Timing Audible Actions via Inflectional Flow"
_ICML.cc/2025/Conference — ICML 2025 poster_

### Official Review · Reviewer_WLLu · 2025-03-10

**Overall Recommendation:** 3

**Summary:**

Authors design a novel task of audible action temporal localization. A new dataset called $Audible623$ and a $TA^2Net$ architecture are proposed for this task. Experiments shows the advantages of them.

**Claims And Evidence:**

Yes.

**Essential References Not Discussed:**

Related works are introduced.

**Experimental Designs Or Analyses:**

The experiment design is suitable.

**Methods And Evaluation Criteria:**

Yes.

**Other Comments Or Suggestions:**

Please refer to weaknesses.

**Other Strengths And Weaknesses:**

**Strengths:**
1. The proposed task of audible action temporal localization is interesting and the motivation is reasonable.
2. The explanation of the task in Figure 1 is clear and vivid.
3. Clear model structure.
4. Reviewer appreciate the experimental results in demo videos.

**Weaknesses:**
1. The cite of Figure 2 precedes Figure 1 in the introduction which is strange. Additionally, the figure name is main text is Figure X instead of Fig.X. Reviewer recommend authors check and apply \figurename~\ref in their manuscript.
2.  Why Figure 2 is not PDF format and fonts within it cannot be selected? Authors are suggested to polish Figure 2.
3. Why authors select Kinetics and UCF101 instead of other widely-used video datasets such as SSv2 and HMDB51?
4. The formula is not properly written, especially the subscripts of the loss. In standard mathematical formulas, the subscripts of such nonvariables should not be used in italics but in regular fonts.
5. The layout of the manuscript should be improved. There are many unreasonable single words that take up an entire line.

**Questions For Authors:**

Please refer to weaknesses.

**Relation To Broader Scientific Literature:**

The task authors proposed is diverges from traditional action recognition and temporal action localization.

**Theoretical Claims:**

The correctness of any proofs for theoretical claims is suitable.

---

> ### Author Rebuttal · Authors · 2025-04-01
>
> **1. Formatting Issues**
>
> We sincerely appreciate your valuable feedback. We will revise the figure citations, adjust the formatting of Figure 2, correct the subscripts in the equations, and improve the overall layout of the paper. Additionally, we will fix the typographical error in line 376 to ensure clarity and consistency throughout the manuscript.
>
> **2. Concerns on Dataset Selection**
>
> We selected Kinetics and UCF101 because they serve as standard benchmarks for action counting, which is a core use case of our task. These datasets include a wide range of repetitive, sound-associated actions such as “clapping”, “jumping jacks”, “typing”, and “punching”, which are well aligned with the objectives of audible action localization.
>
> In contrast, datasets like SSv2 and HMDB51 emphasize abstract or context-dependent interactions that are less relevant to our goal. For example, SSv2 includes many object-centric and non-audible actions such as “pretending to open something” or “moving something from left to right”, which lack consistent visual-sound correspondence. Similarly, HMDB51 features a high proportion of semantically complex or non-repetitive actions such as “smiling” or “listening to music”, which are not suitable for sound grounding or counting tasks.
>
> Therefore, our dataset selection is driven by the need to support visually grounded audible action detection, and we believe it is well justified for the specific focus of our work.

---

### Official Review · Reviewer_kBge · 2025-03-13

**Overall Recommendation:** 3

**Summary:**

The paper proposes a new task named "audible actions" and introduces a new dataset for this task. It introduces an inflectional flow estimation and an auxiliary self-supervised training method to improve the performance. Experiments are conducted on Audible623, UCFRep, and CountixAV datasets.

## update after rebuttal
I keep my initial rating of weak accept after the rebuttal. The rebuttal addressed most of my concerns.

**Claims And Evidence:**

The paper propose "audible actions" as a novel task in TAL. However, it is a subset of the existing TAL task where the actions have audible cues and so existing methods can still work. So, the claim of novelty is not justified.

The proposed method is trained to identify "audible actions". But it can only identify actions that it was trained on. Does it generalize to audible actions beyond the training dataset? There is no discussion on this.

Figure 1 shows a motivating example of audible actions in computer video games. However, visible audible actions are subjective and ambiguous in video games making it difficult to quantify. Further, certain actions such as dribbling a ball can be considered audible or not depending on the decibel level. In essence, the framing of this task is subjective and there is no clear rubric for it.

In real videos, audible actions can occur simultaneously which can result in commotion if there are multiple sources. For example, two persons juggling a ball. How to distinguish between different audio sources in this case?

**Essential References Not Discussed:**

The paper does not cite prior methods which have used optical flow for TAL tasks such as

Dejun Zhang, Linchao He, Zhigang Tu, Shifu Zhang, Fei Han, Boxiong Yang, Learning motion representation for real-time spatio-temporal action localization, Pattern Recognition, Volume 103, 2020.

Yuanzhong Liu, Zhigang Tu, Liyu Lin, Xing Xie, and Qianqing Qin  Real-time Spatio-temporal Action Localization via Learning Motion Representation, ACCV, 2020.

**Experimental Designs Or Analyses:**

Missing quantitative comparisons with popular TAL methods such as BMN and G-TAD.

**Methods And Evaluation Criteria:**

The self-supervised spatial auxiliary training (Sec. 4.2) loss is not very well motivated and it appears to be added just to improve the overall performance.

**Other Comments Or Suggestions:**

Typo - should be "quantitatively" in L376

**Other Strengths And Weaknesses:**

The paper does not discuss the computational complexity of the method, specifically because it uses optical flow  which is expensive.

**Questions For Authors:**

What are guidelines for annotators in choosing audible actions?

How is data annotated for high frequency actions like drumming? Are all frames labeled as audible?

For dataset collection, how does the filtering occur? With action class names? (L152)

How to deal with categories that are not labeled but present in the video? i.e., How does it generalize to novel audible actions?

**Relation To Broader Scientific Literature:**

The key contributions of the paper in introducing the task of predicting audible actions and proposing the model TA2Net. Since, this is a narrow  subset of tasks defined in existing TAL methods, the contribution is limited to a narrow domain that does not benefit the broader TAL community. For example, it would be better if the proposed method can still apply to broader TAL tasks.

**Theoretical Claims:**

No theoretical claims made in the paper.

---

> ### Author Rebuttal · Authors · 2025-04-01
>
> **1. Distinction Between Our Method and TAL Methods, and Applicability to TAL**
>
> We respectfully clarify that our method targets a task that is fundamentally different from traditional TAL. While TAL focuses on localizing the full temporal extent of actions, our method is designed to identify precise keyframes where audible events occur. In this sense, TAL addresses event-level localization, whereas our approach focuses on frame-level detection tied to sound-producing moments.
>
> This distinction is important, as audible frames often represent brief, high-impact instants that are not captured by coarse action intervals. As shown in the appendix, our method also performs well on non-audible actions, demonstrating its ability to capture subtle temporal transitions. We believe this fine-grained capability may provide useful insights for improving temporal precision in future TAL work.
>
> **2. Generalization Beyond the Training Dataset**
>
> Please refer to the reply of Reviewer umcX Q2.
>
> **3. Ambiguity of Audible Action && Multiple Audio Sources**
>
> While the exact decibel level may vary, our task focuses on visually observable actions that are intentionally associated with sound production, consistent with how humans infer sound-related events from visual cues.
> To minimize subjectivity, we annotate dominant and intentional audible actions during training, following standard practice in broader action recognition tasks where annotators focus on salient or representative instances of an action (e.g., annotating the main swing in "golf swing" rather than every subtle motion).
>
> For testing, our model responds to motion patterns indicative of sound, regardless of their intensity. For instance, in the case of basketball dribbling, both soft and forceful dribbles share similar motion dynamics, and our model is designed to detect both.
>
> We also clarify that Fig. 1 is purely for illustration purposes; all samples used in training and evaluation come from real-world, in-the-wild videos with natural variations in appearance and motion.
>
> We will revise the manuscript to better explain our annotation criteria and the illustrative role of Fig. 1.
>
> **4. Multiple Audio Sources**
>
> In scenarios involving multiple audio sources, our method is designed to detect all sound-producing actions present in the video. However, distinguishing between individual sound sources is beyond the current scope of our work. Our primary objective is to accurately identify the timing of audible actions, regardless of their specific source.
>
> **5. Quantitative Comparisons with TAL methods**
>
> We further evaluated BMN on the Audible623 dataset using a two-stream network that incorporates both video and optical flow features. This allows for a fair comparison with TAL methods that also leverage motion information. Since the source code for Zhang et al. and Liu et al. is not publicly available, we have contacted the authors and will include further discussion in the revised manuscript based on their responses.
>
> | Methods | Recall↑ | Precision↑ |  F1↑  |  NME↓  |  PME↓ |
> | :-----: | :-----: | :-----: | :-----: | :-----: | :-----: |
> |  BMN    | 0.417  |   0.486   | 0.413 | 9.327 | 0.951 |
> |  Ours   | 0.648  |   0.656   | 0.616 | 3.462 | 0.744 |
>
> **6. Concerns on Computational Complexity**
>
> Thanks to our lightweight network design, the computational cost of incorporating additional optical flow and inflectional flow features remains manageable. We evaluated the processing time of various methods on the same set of videos. While the inclusion of optical flow does introduce some additional overhead, our method maintains a favorable trade-off and still outperforms most competing approaches in both accuracy and efficiency.
>
> | Methods                        | Time(s) | Methods                     | Time (s)  |
> |--------------------------------|---------|-----------------------------|---------|
> | RepNet                         | 0.075   | Hiera                       | 0.263   |
> | TransRAC                       | 4.301   | ActionFormer                | 0.233   |
> | X3D                            | 0.066   | TriDet                      | 0.240   |
> | TimeSformer                    | 9.968   | Ours (w/ & w/o Flow)    | 0.152 / 0.083 |
>
> **7. Dataset Selection and Annotation**
>
> Annotators were instructed to review each video individually and select visually observable audible actions, excluding videos that either lacked audible actions or featured occluded actions. For high-frequency actions such as drumming, which are particularly challenging to annotate precisely, we adopt a frame-level labeling strategy, marking only the frames where the drumstick makes contact with the drum.
>
> As noted in Sec. 3.2, we do not apply category-based filtering during annotation. Instead, the first round of filtering involves manually reviewing each video to determine whether it contains visually identifiable audible actions.

---

> > ### Comment · Reviewer_kBge · 2025-04-02
> >
> > Thanks to the authors for the rebuttal. It addressed some of my concerns.
> >
> > **Novelty**: The authors mention that proposed method focuses on frame-level detection that produces sounds as compared to TAL methods. However, several TAL methods also output frame-level timestamps (for example, weakly supervised approaches). How is it different when considering the $\textit{visible audible action}$ as a separate category in existing TAL approaches?
> >
> > **Generalization**: The authors refer to the results of generalization to other tasks such as repetitive counting for the question on generalization, however, it is not the same as generalization to new visible audible actions that is not present in the dataset. It would be better to show either qualitative or quantitative results of the proposed method on new videos containing $\textit{unseen audible actions}$.
> >
> > **Quantitative Comparisons with BMN**: What are the features used for BMN? Is it comparable to the proposed $TA^2Net$? The performance of BMN highly depends on the features used as using stronger features such as I3D will obtain better results than weaker features such as 2D TSN.
> >
> > **Complexity**: Is the reported time the average time taken per video?

---

> > > ### Author Response · Authors · 2025-04-02
> > >
> > > **Novelty:**
> > > Thank you for raising this point. While some weakly supervised TAL methods produce frame-level outputs, these are typically used to generate temporal proposals for segment-level localization. Their core objective remains identifying the temporal extent of **predefined action categories** (e.g., "running" or "clapping") based on semantic understanding.
> > >
> > > In contrast, our method is **category-agnostic** and focuses on detecting the precise keyframes where audible events occur, regardless of action class. This distinction brings two key novelties: first, our approach emphasizes low-level motion patterns directly linked to sound production, rather than higher-level semantic labels; second, by not relying on predefined categories, our method can generalize to a wide range of actions, including unseen ones.
> > >
> > > This category-agnostic, keyframe-level detection is particularly well suited for applications such as dubbing and audio-visual synchronization, where temporal precision is more critical than semantic classification.
> > >
> > > **Generalization:**
> > > Our generalization experiments are designed to evaluate how well the model performs on unseen datasets, rather than on new instances within the same distribution. We use UCFRep and CountixAV because they are the only available datasets that are both relevant to our task and completely disjoint from our training data. Our model was never trained on these datasets, making them strong candidates for evaluating cross-dataset generalization.
> > >
> > > The focus on repetitive counting is not because our method is limited to that task, but because these datasets contain repetitive, sound-producing actions that align with the goals of audible action detection. This setting allows us to assess how well our model transfers to different visual content and action instances without retraining.
> > >
> > > Additionally, as shown in our demo, our method generalizes well to animated content, which is visually distinct from all training data. This qualitative result highlights the robustness of our model in detecting sound-producing actions even under substantial domain shifts. We appreciate the suggestion and agree that further testing on newly collected audible action categories would provide additional support, which we plan to explore in future work.
> > >
> > > **Quantitative Comparisons with BMN:**
> > > To ensure fairness, we use the same frame-level visual and optical flow features, generated by our encoder, for both our method and BMN. We intentionally avoid using I3D features, as they are tailored for snippet-level representations and performed worse in preliminary experiments for fine-grained localization. By keeping the features consistent, we isolate the modeling performance and ensure a direct comparison.
> > >
> > > **Complexity:**
> > > Yes, the reported inference time represents the average per video on the test set, where each video contains approximately 250 frames. This provides a clear and consistent basis for comparing computational efficiency across methods.

---

### Official Review · Reviewer_umcX · 2025-03-13

**Overall Recommendation:** 3

**Summary:**

This paper introduces a new task called audible action temporal localization, aimed at predicting the frame-level positions of visible audible actions.

And the paper further proposes a dedicated dataset called Audible623, derived from Kinetics and UCF101.

Finally, the paper proposes a baseline method TA$^2$Net which employs flow estimation based on motion's second derivative.

## update after rebuttal
I keep my initial rating of weak accept. The rebuttal addressed some of my concerns.

**Claims And Evidence:**

1.	The authors introduce the task of Audible Action Temporal Localization, aimed at pinpointing the spatio-temporal coordinates of audible movements. However, there are no spatial annotations in the proposed dataset. The authors propose an auxiliary training method that leverages spatial information and produces a localization map as a side-output.

2.	The authors claim that the difference between Temporal Action Analysis and Audible Action Temporal Localization is that one focuses on event-level localization, while the other focuses on accurate key frame identification. And the authors formulate the task as aiming to determine whether each frame contains action that can generate sound. However, the timing boundaries of certain action in the video inevitably exhibit some degree of ambiguity. The authors have not discussed this ambiguity issue in the Dataset Annotation section.

3.	The proposed dataset is derived from Kinetics and UCF101 which contains at least 400 and 101 action classes respectively. And the proposed dataset contains only 14 categories of audible actions. The reviewer has concerns about the generalizability of the proposed dataset and its applicability in helping the dubbing task.

**Essential References Not Discussed:**

N/A

**Experimental Designs Or Analyses:**

N/A

**Methods And Evaluation Criteria:**

N/A

**Other Comments Or Suggestions:**

1.	Typos in the caption of Figure 4 (Page4-Line181).

**Other Strengths And Weaknesses:**

Strengths

1. Proposing a new task, constructing a dedicated dataset and proposing a baseline method with comprehensive comparisons requires a significant amount of effort. And the paper is written in a relatively clear and readable manner.

Weaknesses

1. The definition of visible audible action remains ambiguous. And judging from the proposed dataset, the audible action is merely a small subset of the existing action categories.

**Questions For Authors:**

Since the proposed new task is close to the task of traditional Spatial Temporal Action Localization/Temporal Action Detection, why would the authors construct the new benchmark using the public datasets on these tasks? It seems the temporal annotation would be easier and the spatial-temporal annotations can be reused.

**Relation To Broader Scientific Literature:**

N/A

**Theoretical Claims:**

N/A

---

> ### Author Rebuttal · Authors · 2025-04-01
>
> **1. Ambiguity on Timing Boundaries of Actions**
>
> Temporal boundary ambiguity is indeed a well-known and widely acknowledged challenge in general action understanding tasks. Precisely defining the start and end of an action, particularly for semantically complex or continuous activities such as "sit down" or "throw the ball", is inherently difficult due to visual similarity across adjacent frames and the subjective nature of temporal segmentation.
>
> However, unlike general action semantic understanding, our task, Audible Action Temporal Localization, is significantly more constrained and objective. Rather than requiring full temporal segmentation of action intervals, we focus on identifying the specific frames in which a sound is produced by an action. These "sounding frames" are typically easier to annotate consistently, as the accompanying audio provides a clear and reliable temporal anchor.
> We will clarify this annotation protocol and emphasize the inherent simplicity and objectivity of our task in the revised Dataset Annotation section.
>
> **2. Concerns on Dataset Categories**
>
> Our task is fundamentally different from the original objectives of Kinetics and UCF101. While those datasets are designed for action classification across a broad range of semantic categories, our focus is category-agnostic, targeting the detection of action patterns rather than semantic understanding. As a result, the Audible623 dataset does not include category annotations but instead provides precise temporal labels for frames where audible events occur. This design is well aligned with the goals of dubbing and audio-visual synchronization, which rely on accurate temporal localization rather than action recognition.
>
> Thanks to the design of our framework, our method demonstrates strong generalization ability. As shown in Table 3, it performs well on datasets with different action categories, such as UCFRep and CountixAV. This confirms that our approach is not constrained by specific categories and can generalize effectively to unseen actions. For instance, our method was successfully applied to animated content in the demo without any additional training.
>
> We will clarify these distinctions and elaborate on the rationale behind our dataset design in the revised manuscript.
>
> **3. Audible Action Definition**
>
> While audible actions are a subset of general actions, they serve a distinct and crucial role in applications like sound dubbing and repetitive counting. Unlike general action recognition, which often relies on high-level and potentially ambiguous semantic categories (e.g., "playing sports," "interacting"), audible actions are grounded in concrete, low-level visual cues that correlate directly with sound production (e.g., clapping, stomping, tapping). This distinction makes it category-agnostic and more robust to domain shifts. This targeted focus enables reliable grounding of actions with sound, which is essential for many practical tasks.
>
> **4. Reuse of TAL Datasets**
>
> As mentioned in Section 3.2, existing Temporal Action Localization (TAL) datasets are not directly applicable to our task. The keyframe-level labels required for Audible Action Temporal Localization cannot be reliably derived from the start and end frames of action segments in these datasets. In many cases, the annotations are relatively coarse and do not separate repeated instances of the same action. For example, a kicking sequence with multiple discrete kicks is often labeled as a single continuous segment, whereas our task focuses on detecting each individual audible event.
>
> Additionally, TAL datasets include many inaudible actions that fall outside the scope of our task. Reusing these datasets would require significant re-annotation and filtering to align with our objective. For these reasons, we chose to construct a dedicated benchmark with precise frame-level annotations that better support the goals of audible action localization and audio-visual synchronization.

---

### Official Review · Reviewer_v1Hi · 2025-03-15

**Overall Recommendation:** 3

**Summary:**

This paper introduces the task of audible action temporal localization and proposes a novel framework $TA^2Net$ alongside the Audible623 dataset. The method features a inflectional flow estimation technique grounded in the second derivative of the position-time images. Additionally, the authors develop a self-supervised spatial localization. The effectiveness of the proposed method is demonstrated, and its broad applicability is validated in other tasks.

**Claims And Evidence:**

please refer to Weaknesses.

**Essential References Not Discussed:**

please refer to Weaknesses and Questions For Authors.

**Experimental Designs Or Analyses:**

I did. The experimental design and analysis are sound and valid. The authors have provided.

**Methods And Evaluation Criteria:**

While the proposed dataset demonstrates partial validity for audible action temporal localization, its ability to capture real-world complexity remains questionable. The current evaluation primarily focuses on short-duration, scripted scenarios (e.g., 9.2-second clips in Audible623) with controlled audio-visual correspondence.

**Other Comments Or Suggestions:**

please refer to Weaknesses.

**Other Strengths And Weaknesses:**

### Strengths

1. **Novel Task FormulationL:** Proposes the first formal task of audible action temporal localization.
2. **Rigorous Validation：** Demonstrates broader applicability through:
    * Cross-task generalization experiments
    * Vision-language model (VLM) evaluations in supplements

### Weaknesses

1. **Lack of user study:** A potential issue with $TA^2Net$ is that it shows slightly weaker performance on certain metrics compared to some baselines for the audible action temporal localization task, which may make the subjective human opinions of higher importance to the present manuscript. This should further validate how well the paper's core assumption aligns with the actual human preferences.
2. **Mismatch Between Data Characteristics and Claimed Application Scenarios:** The proposed method is primarily evaluated on short and low-framerate videos (e.g., Audible623 dataset, with an average duration of 9.2 seconds and 250 frames per video). While this setup may suffice for initial validation, it significantly limits the generalizability of the method to real-world applications. In practice, videos on platforms like YouTube, TikTok, or in professional contexts (e.g., movies, TV dramas) are typically longer and have higher framerates, requiring more robust temporal modeling and scalability. The current experiments do not adequately address these challenges.
3. **Methodological Ambiguity in Motion Feature Extraction:** The paper’s reliance on Xu et al.'s pre-trained optical flow network raises concerns about feature selection rationale. While the chosen network can extract multi-modal motion features (e.g., depth, disparity), the authors exclusively utilize optical flow without justifying why supplementary features were disregarded. This omission is particularly notable given that depth/disparity information could enhance temporal modeling in complex scenes. Furthermore, the decision to adopt Xu et al.'s architecture over established alternatives like FlowNet or PWC-Net lacks explicit justification. Critical factors such as computational efficiency (e.g., inference speed comparisons), benchmark performance, or inherent architectural advantages for audible action temporal localization tasks remain unaddressed. These unresolved design choices cast doubt on whether the selected framework optimally balances accuracy and practicality. To strengthen the methodology, the authors should either provide empirical evidence or theoretical arguments explaining why optical flow alone suffices for their task.

**Questions For Authors:**

The paper computes forward optical flow between consecutive frames using a pre-trained network (Xu et al., 2023). Could the authors clarify why this approach was prioritized over classical optical flow algorithms (e.g., DisFlow, Brox Flow) that are widely adopted in video processing?

**Relation To Broader Scientific Literature:**

please refer to Weaknesses and Questions For Authors.

**Theoretical Claims:**

I have reviewed the theoretical claims in Sections 4.1 (Timing Audible Actions with Inflectional Flow) and 4.2 (Self-supervised Spatial Auxiliary Training). While the presented formulations are logically consistent and free of apparent errors, their relative simplicity warrants discussion. The derivations primarily involve elementary operations without addressing more complex scenarios (e.g., non-linear motion patterns).

---

> ### Author Rebuttal · Authors · 2025-04-01
>
> **1. User Study**
>
> As suggested, we conducted a user study comparing five approaches: TriDet, TransRAC, X3D, Hiera, and our proposed method. We curated a set of eight videos and enlisted 30 participants for the study. Each video was dubbed using sound aligned to the audible action locations detected by each method. Participants were then asked to assess the audio-visual synchronization quality and select the top two methods they perceived as most accurate for each video. The results show that our method was most frequently selected as the best, indicating a clear preference and superior perceived synchronization performance.
>
> | Method   | Top-2 Rate|
> |----------|----------|
> | Ours     | 0.8458 |
> | TriDet   | 0.2208 |
> | TransRAC | 0.5625 |
> | X3D      | 0.1125 |
> | Hiera    | 0.2583 |
>
> **2. Concerns on Short Duration & Application Scenarios**
>
> We respectfully disagree with this concern, as it appears to stem from a misunderstanding of both the dataset characteristics and our method’s design. The video duration and frame rate in the Audible623 dataset, as well as in other benchmarks used in our study, are consistent with established standards in the action counting and localization community. These datasets are widely adopted to ensure fair comparisons and reproducibility. More importantly, our method operates at the frame level, making it inherently agnostic to the overall video length. This design choice allows the method to scale effectively to longer and higher-framerate videos without degradation in localization performance. As such, our approach remains robust and applicable to real-world scenarios, including platforms like YouTube and professional media content.
>
> **3. Methodological Ambiguity in Motion Feature Extraction**
>
> We respectfully believe this concern may stem from a misunderstanding of motion feature roles. Depth and disparity capture static spatial structure but do not reflect temporal dynamics. For example, counting the number of times a person jumps depends on detecting repeated vertical motion—something depth alone cannot convey. In contrast, optical flow reflects dynamic changes across frames and is widely recognized as a reliable signal for temporal modeling.
>
> While depth can theoretically be used to extend 2D optical flow to 3D, such usage is rare in practice, as action scenarios that truly benefit from 3D motion modeling are uncommon and often domain-specific. Moreover, incorporating depth adds computational overhead without clear gains for typical action tasks.
>
> We chose Xu et al.’s network for its solid empirical performance and a good trade-off between accuracy and efficiency. While alternatives like FlowNet or PWC-Net are well-known, we did not observe significant advantages for our task. We will clarify this rationale in the revised manuscript.
>
>
> **4. Optical Flow Method Selection**
>
> Compared to traditional methods such as DisFlow, modern neural network-based approaches offer significantly improved robustness in challenging real-world scenarios involving occlusion, noise, or complex motion, and they perform consistently well across varying resolutions. Among these, GMFlow demonstrates state-of-the-art accuracy while maintaining inference times comparable to established methods like PWC-Net [1, 2]. Additionally, GMFlow offers a more streamlined and efficient framework for computing bidirectional optical flow, which is beneficial for our application. Based on these advantages and after thorough consideration, we selected GMFlow for optical flow extraction in our pipeline.
> [1] Xu H, Zhang J, Cai J, et al. Gmflow: Learning optical flow via global matching, CVPR, 2022.
> [2] Teed Z, Deng J. Raft: Recurrent all-pairs field transforms for optical flow, ECCV, 2020.
>
> **Note to Reviewer**
>
> We appreciate the reviewer’s time and effort in evaluating our work. While we find that some of the comments may stem from misunderstandings of our method and its design choices, we have addressed each point in detail and clarified the rationale behind our decisions. We believe these clarifications resolve the concerns raised and further highlight the strengths and contributions of our approach. We respectfully ask the reviewer to reconsider the insights and potential impact of our work in light of these responses.

---

> > ### Comment · Reviewer_v1Hi · 2025-04-05
> >
> > thanks for rebuttal.  While I have read the rebuttal and all the comments of other reviewers, the concerns are still maintained. I will keep the rating.

---

> > > ### Author Response · Authors · 2025-04-05
> > >
> > > We believe we have thoroughly and carefully addressed the reviewer’s concerns. In response, we conducted a user study directly comparing our method against multiple baselines, offering empirical validation of its effectiveness. We also clarified the misunderstanding regarding video duration and application scenarios, provided a detailed explanation for our use of optical flow—highlighting its unique role in modeling temporal dynamics—and justified why depth or disparity are not suitable for capturing sound-related motion.
> > >
> > > Furthermore, we offered a clear rationale for selecting GMFlow, supported by discussion on its balance of accuracy and efficiency, and evaluated the impact of incorporating optical flow on time complexity.
> > >
> > > Given these clarifications and additional results, we hope the reviewer can engage further with the specific points raised, as simply stating that concerns remain—without responding to the evidence or explanation provided—does not support a constructive or professional review process. We remain open to detailed feedback that will help further refine our work.

---

### Decision · Program_Chairs · 2025-05-01

**Decision:**

Accept (poster)

**Comment:**

The paper proposes a new task of audible action temporal localization and a new dataset for this task, namely Audible623. Different from traditional Temporal Action Localization (TAL) task, where the model is tasked to localize (temporally) instances of a set of pre-defined actions. Audible TAL is formulated as detecting moments (keyframes) from the videos that make sound. The paper also presents a novel approach with solid experiments.

After rebuttal, most of concerns raised by reviewers are addressed by the authors. And all reviewers recommend weak accept. AC reads all reviews, rebuttal and discussions and agrees with the reviewers, thus recommends a weak accept.

A few minor comments:
  * The dataset is quite small ~ 600 videos which may make the findings less convincing.
  * Missing a few references: [1] Zhao et al., The Sound of Pixels, ECCV 2018, [2] Alwassel et al., Self-Supervised Learning by Cross-Modal Audio-Video Clustering, NeurIPS 2020.